# UP-NeRF: Unconstrained Pose-Prior-Free Neural Radiance Fields

**Injae Kim**[*]
Korea University
dna9041@korea.ac.kr

**Minhyuk Choi**[*]
Korea University
sodlqnf123@korea.ac.kr

**Hyunwoo J. Kim**[†]
Korea University
hyunwoojkim@korea.ac.kr

## Abstract

Neural Radiance Field (NeRF) has enabled novel view synthesis with high fidelity given images and camera poses. Subsequent works even succeeded in eliminating the necessity of pose priors by jointly optimizing NeRF and camera pose. However, these works are limited to relatively simple settings such as photometrically consistent and occluder-free image collections or a sequence of images from a video. So they have difficulty handling unconstrained images with varying illumination and transient occluders. In this paper, we propose **UP-NeRF** (**U**nconstrained **P**ose-prior-free **Ne**ural **R**adiance **F**ields) to optimize NeRF with unconstrained image collections without camera pose prior. We tackle these challenges with surrogate tasks that optimize color-insensitive feature fields and a separate module for transient occluders to block their influence on pose estimation. In addition, we introduce a candidate head to enable more robust pose estimation and transient-aware depth supervision to minimize the effect of incorrect prior. Our experiments verify the superior performance of our method compared to the baselines including BARF and its variants in a challenging internet photo collection, *Phototourism* dataset. The code of UP-NeRF is available at `https://github.com/mlvlab/UP-NeRF`.

## 1 Introduction

Neural Radiance Fields (NeRF) [1] opened up a new chapter in novel view synthesis by demonstrating its capacity to generate high-quality novel view images given only a set of 2D images. Its powerful performance has enabled many practical applications including virtual/augmented reality (VR/AR) [2, 3], autonomous systems [4, 5], and robotics [6–8]. Various follow-up studies have developed NeRF by addressing its limitations, such as the necessity of dense input views [9–11] and long training time [12–17]. Also, there have been efforts [18, 19] to deal with unconstrained images with varying illuminations and transient occluders to relieve the burden of collecting clean images or establishing a specialized setting to capture them. For instance, NeRF-W [18] introduces an appearance embedding to handle varying illumination and a transient head to filter transient occluders.

One of the popular research topics in NeRF is joint optimization for pose and neural scene representation. Originally, NeRF requires accurate camera pose priors which are generally obtained with classic methods like structure from motion (SfM) [20]. Although the poses obtained from such a method are treated as ground truth in many works, it does not always give optimal results or even fails to converge. Hence recent works [21–23] have proposed a method so-called *unposed-NeRF* that simultaneously trains camera poses and NeRF. This eliminates the necessity of pose priors and burdensome preprocessing. While these methods have only succeeded in forward-facing scenes, NoPe-NeRF [24] further improved the method to even outdoor scenes with continuous images from videos. These previous works, however, are based on the photometric loss which is not reliable for

---

[*]First two authors have an equal contribution.
[†]Corresponding author.

37th Conference on Neural Information Processing Systems (NeurIPS 2023).

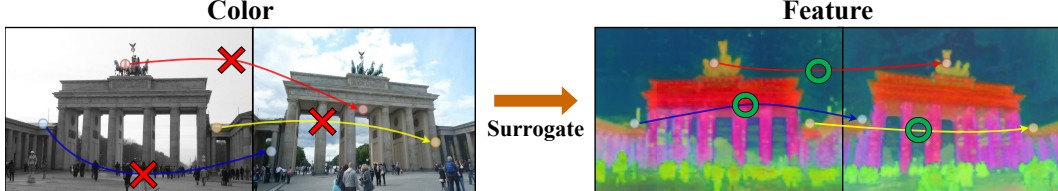

Figure 1: Deep feature focuses on semantic information of objects rather than their colors, which makes it an effective descriptor for images with photometric inconsistency.

unconstrained images, so they have difficulty in training with unconstrained images. In addition, the unconstrained images have complex camera poses, which is hard to optimize if a prior like continuity of images from a video is not available. Therefore, training unposed-NeRF with unconstrained images causes contradiction: To estimate camera poses accurately, images must be photometrically consistent without transient occluders; to treat unconstrained images without photometric consistency and transient occluders, accurate camera poses are needed. Previous unposed-NeRFs in the literature are not capable of learning in an unconstrained setting, leading to suboptimal pose estimation and poor rendering quality.

To tackle the problem, we propose **UP-NeRF** (**U**nconstrained **P**ose-prior-free **Ne**ural **R**adiance **F**ield) that jointly optimizes pose and neural scene representations while minimizing the influence of photometric inconsistency and transient occluders on pose estimation. Our contributions are as follows: First, we introduce the novel architecture that enables robust pose optimization with images of complex camera poses. Second, we propose feature-surrogate bundle adjustment by adopting the deep feature as a descriptor. Third, we enable transient-free pose optimization with the isolated network. Lastly, our transient-aware depth loss suggests an effective way to impose depth prior only to the static objects. Consequently, our model enables successful training of unposed-NeRF with unconstrained images, showing its effectiveness over a naive combination of current methods.

## 2 Related Work

**Neural Radiance Fields (NeRF).** With the advent of techniques adopting neural fields to reconstruct 3D representations [25–29], Neural Radiance Field (NeRF) [1] proposed by Mildenhall *et al.* showed a promising result without the requirement of exact 3D geometry of each object in the scene. Compared to previous approaches which require a pre-defined representation of the scene by meshes [30, 31], voxels [25, 26], or point clouds [27, 32], NeRF only requires a set of densely captured 2D images and corresponding camera parameters of each image. Despite its light requirements, it demonstrates quality reconstruction of the complex geometry of scenes. The key idea of NeRF is that even though given are only 2D images, an implicit function represented by a multilayer perceptron can be utilized to parameterize the color and density of each 3D point on a ray, which is subsequently synthesized to decide the color of a pixel from which the ray was shot. Thanks to its simple but effective architecture, many other researches [10, 12–14, 33–35] have followed the trail of NeRF to ameliorate its weakness and improve performance.

**Joint optimization of camera pose.** Among a lot of interesting NeRF-based works, one major group of following works deals with the requirement of accurate camera pose of each image. Accurate camera pose including intrinsic and extrinsic parameters of a camera is indispensable for NeRF in that sampling accurate 3D coordinates in the world space heavily depends on them. However, such an accurate camera pose is not easily equipped unless the dataset is custom-made like a synthetic dataset, so structure from motion (SfM) [36] has been a classical solution to obtain such parameters. Although a lot of works have postulated camera poses from SfM to be ground truth, they can be sub-optimal depending on properties of scenes like monotonous texture. To solve this problem, other methods [22–24, 37–40] propose joint optimization of a neural field and poses.

**Expansion to unconstrained scenes.** Several researches [18, 19] dealt with unconstrained outdoor images where color consistency is not guaranteed owing to various environments (sunshine, time, weather, etc.) and so-called *transient* entities (pedestrian, cars, sundries, etc.) appear in images.

S-NeRF [41] and CF-NeRF [42] proposed a Bayesian network and normalizing flow for measuring the uncertainty of transient objects. $D^2$NeRF [43] and RobustNeRF [44] demonstrated that transient objects can be sharply decoupled from static ones without blurry floaters. Some works [45, 46] utilized mask supervision to jointly optimize camera pose with object-centric images of different illumination. In contrast to previous works which require either color consistency or the absence of transient objects to jointly optimize the pose, our method is capable of optimizing camera pose without both requirements.

## 3   Method

We present a framework dubbed UP-NeRF to optimize unposed-NeRF under challenging conditions. We first briefly summarize Neural Radiance Fields (NeRF) [1] followed by joint optimization of camera poses and NeRF. In Section 3.1, we propose a new architecture that handles different convergence rates of camera poses, which can enable robust training of unposed-NeRF. Next, we explain our methods in Section 3.2 and 3.3 that handle the difficulty of unconstrained image collections like different light conditions (*e.g.*, weather, time, etc.) and transient occluders. In addition, we will discuss how monocular depth priors can be used to optimize unposed-NeRF with unconstrained images in Section 3.4.

**Neural Radiance Fields (NeRF).**   Given $N$ images $\{\mathbf{I}_i\}_{i=1}^{N}$ and camera parameters, Neural Radiance Field (NeRF) $\Pi_\theta$ learns a continuous volumetric radiance field which maps a 3D point $\mathbf{x} = (x, y, z)$ and a viewing direction $\mathbf{d} = (d_x, d_y, d_z)$ to volume density $\sigma$ and color $\mathbf{c} = (r, g, b)$. The mapping functions parameterized by simple neural networks, *e.g.*, multi-layer perceptrons (MLPs), are defined as

$$[\sigma(t), \mathbf{z}(t)] = \Pi_{\theta_1}\big(\gamma_\mathbf{x}(\mathbf{r}(t))\big), \quad \mathbf{c}(t) = \Pi_{\theta_2}\big(\mathbf{z}(t), \gamma_\mathbf{d}(\mathbf{d})\big), \tag{1}$$

where $\gamma_\mathbf{x}(\cdot)$ is a positional encoding function for 3D coordinates of each sample point on the ray $\mathbf{r}(t) = \mathbf{o} + t\mathbf{d}$, which is parameterized by the origin point $\mathbf{o}$ and the direction $\mathbf{d}$. Color value $\mathbf{c}(t)$ is encoded with an intermediate feature $\mathbf{z}(t)$ and a positional encoding function for view directions $\gamma_\mathbf{d}(\cdot)$. The synthesized RGB value $\hat{\mathbf{C}}(\mathbf{r})$ is calculated by volume rendering as

$$\hat{\mathbf{C}}(\mathbf{r}) = \sum_{k=1}^{K} T(t_k)\alpha(\sigma(t_k)\delta_k)\mathbf{c}(t_k),$$

$$\text{where } T(t_k) = \prod_{k'=1}^{k-1} (1 - \alpha(t_{k'})) = \exp\left(-\sum_{k'=1}^{k-1} \sigma(t_{k'})\delta_{k'}\right), \tag{2}$$

where $\delta_k = t_{k+1} - t_k$ is the distance between adjacent points for approximated rendering via quadrature on discrete points and $\alpha(t_k) = 1 - \exp(-\sigma(t_k)\delta_k)$. To improve sampling efficiency, NeRF employs coarse and fine models for hierarchical sampling. The coarse model is optimized using stratified sampling, and then the fine model is optimized with samples biased toward the relevant parts of the volume. The parameters are optimized by minimizing photometric loss $\mathcal{L}_{\text{rgb}} = \sum \|\mathbf{C}(\mathbf{r}) - \hat{\mathbf{C}}(\mathbf{r})\|_2^2$ with given ground-truth color $\mathbf{C}(\mathbf{r})$. Note that even if it is not a color map, any information that represents the scene can be used for training a neural field. In this paper, we exploit deep feature map $\mathbf{F}$ as well as color map $\mathbf{C}$ for training, which have been studied by Kobayashi et al. [47] to learn selectively editable neural fields.

**NeRF with camera pose estimation.**   Some prior works optimize NeRF with a joint estimation of camera pose. We parameterize the camera poses $\{\mathbf{p}_i\}_{i=1}^{N}$ as 6 degrees of freedom and assume that the camera intrinsics are known as several prior works [23, 24, 39, 48]. Then, the problem can be formulated as

$$\theta^*, \mathbf{p}^* = \arg\min_{\theta, \hat{\mathbf{p}}} \mathcal{L}_{\text{rgb}}(\hat{\mathbf{C}}, \hat{\mathbf{p}}|\mathbf{C}), \tag{3}$$

where $\hat{\mathbf{p}}$ is learnable camera pose. BARF [23] shows that a coarse-to-fine positional encoding strategy facilitates more stable joint optimization of NeRF and camera pose. Following BARF, we parameterize camera poses with the $\mathfrak{se}(3)$ Lie algebra and use a coarse-to-fine strategy for pose optimization.

## 3.1 Candidate head for robust pose optimization

Training unposed-NeRF is a joint optimization process with a chick-and-egg problem; estimation of camera pose $\hat{\mathbf{p}}$ requires accurate scene representation $\Pi_\theta$ and vice versa. During the initial phase of joint training, we observed that inaccurate scene representation, which is represented by predicted density $\sigma$ and color $\mathbf{c}$ in Eq. (1), causes specific images to be optimized in the wrong poses. We refer to these images as 'hard-pose' images, which may include images that capture the details of the scene or have little overlap with other images like the image in Fig. 4. To mitigate this problem, we introduce an additional MLP head called *candidate head*. It yields image-dependent (and tentative) scene representations by density $\sigma_i^{(c)}$ and color $\mathbf{c}_i^{(c)}$ given candidate embeddings $\{\ell_i^{(c)}\}_{i=1}^N$ as:

$$[\sigma_i^{(c)}(t),\ \mathbf{c}_i^{(c)}(t)] = \Pi_{\theta_3}\big(\mathbf{z}(t), \ell_i^{(c)}\big). \tag{4}$$

Note that $(\cdot)^{(c)}$ denotes that it is related to the candidate embeddings. The synthesized RGB value $\hat{\mathbf{C}}_i^{(c)}$ is generated by a joint volumetric rendering of the shared representation ($\sigma$ and $\mathbf{c}$) and the image dependent representation ($\sigma_i^{(c)}$ and color $\mathbf{c}_i^{(c)}$):

$$\hat{\mathbf{C}}_i^{(c)}(\mathbf{r}) = \sum_{k=1}^K T_i^{(c)}(t_k)\bigg(\alpha(\sigma(t_k)\delta_k)\mathbf{c}(t_k) + \alpha(\sigma_i^{(c)}(t_k)\delta_k)\mathbf{c}_i^{(c)}(t_k)\bigg),$$
$$\text{where } T_i^{(c)}(t_k) = \exp\bigg(-\sum_{k'=1}^{k-1}\big(\sigma(t_{k'}) + \sigma_i^{(c)}(t_{k'})\big)\delta_{k'}\bigg). \tag{5}$$

The intuition is to consider easy pieces first to complete a large jigsaw puzzle and handle hard pieces later. In the initial stages of training, shared representation is predominantly learned by 'easy-pose' images. Meanwhile, candidate representation is primarily trained by hard-pose images. This remedy effectively prevents the shared representation from being distracted by the wrong supervision introduced by hard-pose images.

As the shared representation becomes accurate enough to facilitate the pose estimation of hard-pose images, the influence of the candidate head is gradually reduced so that hard-pose images can be assimilated into the shared representation, and the final model uses only the shared representation, $\sigma$ and $c$. To achieve that, we propose a loss scheduling given as:

$$\mathcal{L} = w_{u,v}\ \mathcal{L}_{\text{rgb}}(\hat{\mathbf{C}}|\mathbf{C}) + (1 - w_{u,v})\ \mathcal{L}_{\text{rgb}}(\hat{\mathbf{C}}_i^{(c)}|\mathbf{C}), \tag{6}$$

where $w_{u,v}$ is defined as:

$$w_{u,v}(l) = \begin{cases} 0 & \text{if } l < u \\ \dfrac{1 - \cos(\pi(l-u)/(v-u))}{2} & \text{if } u \leq l < v \\ 1 & \text{if } l \geq v. \end{cases} \tag{7}$$

Here $l \in [0,1]$ denotes training progress and $u, v \in [0,1]$ are hyperparameters. Then, $w_{u,v}$ tunes the influence of $\hat{\mathbf{C}}_i^{(c)}$ and $\hat{\mathbf{C}}$. Specifically, for learning during the initial stage of training ($l < u$), only $\hat{\mathbf{C}}_i^{(c)}$ is utilized, whereas only $\hat{\mathbf{C}}$ is employed in the later stage ($l > v$). With the loss scheduling, a high-quality final NeRF model can be trained. In our final training pipeline, we will use features instead of colors, as shown in Fig. 2, which will be covered in Section 3.2.

The volume rendering method shown in Eq. (5) is similar to NeRF-W [18] in that it has an additional head for color and density. In our architecture, however, we do not have a regularization term and uncertainty field that hinder the candidate head from handling hard-pose images. The size of embedding $\ell_i^{(c)}$ is adjusted by how challenging the poses of the given images are, which can be viewed as adjusting the regularizers for 'slack' variables in the constrained optimization problem, *e.g.*, support vector machine (SVM). The quantitative analysis of the candidate embedding size is provided in Table 4.

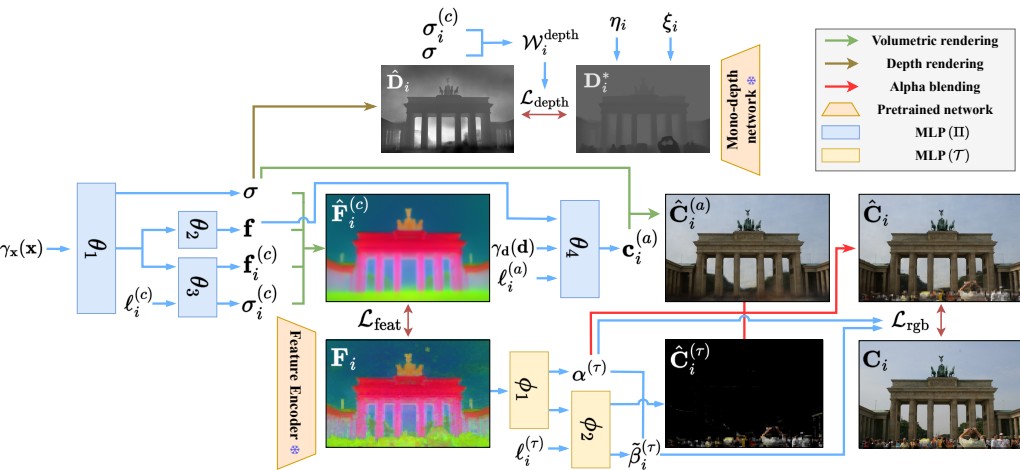

Figure 2: Overall training pipeline of our UP-NeRF ($\Pi_\theta$ is based on Section 3.2). To learn the poses robustly, image-dependent scene representation is learned by the candidate head $\Pi_{\theta_3}$ (Section 3.1). The depth loss $\mathcal{L}_{\text{depth}}$ provides transient-aware depth prior to the model (Section 3.4). In the early stages of training, the deep feature $\mathbf{F}$ is used for color-independent surrogate optimization (Section 3.2). After the poses are roughly learned, the NeRF is optimized with a separated TransientNet $\mathcal{T}_\phi$ to prevent unnecessary gradient flowing the pose parameters (Section 3.3).

## 3.2    Feature-surrogate bundle adjustment

The motivation of this section is shown in Fig. 1. Given images with diverse appearances due to different weather and times, unposed-NeRF cannot be stably trained with a photometric loss since RGB values of corresponding pixels between images are not guaranteed to be the same. We adopt the deep features from ViT [49] as a surrogate, which are less sensitive to appearance changes, and we train the neural field with these features to achieve better robustness under challenging conditions.

With a slight change in Eq. (1), our feature field is defined as:

$$\mathbf{f}(t) = \Pi_{\theta_2}\big(\mathbf{z}(t)\big),$$
$$[\,\mathbf{f}_i^{(c)}(t), \sigma_i^{(c)}(t)] = \Pi_{\theta_3}\big(\mathbf{z}(t), \ell_i^{(c)}\big). \tag{8}$$

Unlike the color $\mathbf{c}(t)$ encoding in Eq. (1), the feature $\mathbf{f}(t)$ encoding is performed without a viewing direction because the features ought to be independent of viewing directions. Furthermore, $\hat{\mathbf{F}}_i^{(c)}(\mathbf{r})$ is rendered similarly to Eq. (5) by volume rendering with $\mathbf{f}(t)$, $\sigma(t)$, $\mathbf{f}_i^{(c)}(t)$ and $\sigma_i^{(c)}(t)$. It is then optimized by minimizing $\mathcal{L}_{\text{feat}}(\mathbf{r}) = \|\mathbf{F}_i(\mathbf{r}) - \hat{\mathbf{F}}_i^{(c)}(\mathbf{r})\|_2^2$ with 2D deep feature $\mathbf{F}_i(\mathbf{r})$ from a pretrained model. We replace the loss $\mathcal{L}_{\text{rgb}}(\hat{\mathbf{C}}_i^{(c)}|\mathbf{C}_i)$ in Eq. (6) with the loss $\mathcal{L}_{\text{feat}}(\hat{\mathbf{F}}_i^{(c)}|\mathbf{F}_i)$ for the early training.

Meanwhile, the unconstrained images can have different colors even though they have the same semantic object. As a result, the feature $\mathbf{f}(t)$ can be used to encode image-dependent static color $\mathbf{c}_i^{(a)}$ along with viewing direction $\mathbf{d}_i$ and appearance embedding $\{\ell_i^{(a)}\}_{i=1}^N$ for each image as:

$$\mathbf{c}_i^{(a)}(t) = \Pi_{\theta_4}\big(\mathbf{f}(t), \gamma_\mathbf{d}(\mathbf{d}), \ell_i^{(a)}\big). \tag{9}$$

Note that $(.)^{(a)}$ means it is related to the appearance embeddings. The predicted RGB value $\hat{\mathbf{C}}_i^{(a)}(\mathbf{r})$ is calculated by volume rendering Eq. (2) with $\mathbf{c}_i^{(a)}(t)$ and $\sigma(t)$. This static color $\hat{\mathbf{C}}_i^{(a)}(\mathbf{r})$ is optimized jointly with the transient color $\hat{\mathbf{C}}_i^{(\tau)}(\mathbf{r})$ that will be introduced in the next section, Section 3.3. In summary, through candidate scheduling Eq. (6), the feature field is trained only for the early stage ($l < u$), $\Pi_{\theta_3}$ is not involved in learning in the later stage of training ($l \geq v$), and the radiance field starts training as feature field gradually stops training ($u \leq l < v$). See Fig. 2 for the final NeRF model $\Pi_\theta$ architecture.

## 3.3 Isolated network for transient-free pose optimization

The next challenge is the occlusion of transient objects. To optimize the poses accurately, their influence must be minimized in the training process as much as possible. The previous work NeRF-W [18], which tackles this problem, models it in 3D space by adding a transient head. If we naively adopt it in our model, it causes a noisy gradient flowing into the learnable pose parameters, with the pose being learned in the wrong direction due to transient occluders. So we mitigate this problem by handling transients occluders at the separated *TransientNet* $\mathcal{T}_\phi$ in 2D image level. It maps deep feature $\mathbf{F}_i(\mathbf{r})$ to opacity $\alpha^{(\tau)}$, which is intended to filter out the transient based on semantic information. Then, along with transient embedding $\{\ell_i^{(\tau)}\}_{i=1}^N$, it encodes image-dependent transient color $\hat{\mathbf{C}}_i^{(\tau)}$ and uncertainty $\tilde{\beta}_i^{(\tau)}$:

$$
\begin{aligned}
{[}\alpha^{(\tau)}(\mathbf{r}), \mathbf{z}^{(\tau)}(\mathbf{r})] &= \mathcal{T}_{\phi_1}\big(\mathbf{F}_i(\mathbf{r})\big), \\
{[}\hat{\mathbf{C}}_i^{(\tau)}(\mathbf{r}), \tilde{\beta}_i^{(\tau)}(\mathbf{r})] &= \mathcal{T}_{\phi_2}\big(\mathbf{z}^{(\tau)}(\mathbf{r}), \ell_i^{(\tau)}\big).
\end{aligned}
\tag{10}
$$

Note that $(.)^{(\tau)}$ means it is related to the transient objects. The final predicted color $\hat{\mathbf{C}}_i(\mathbf{r})$ is obtained by alpha blending with $\hat{\mathbf{C}}_i^{(a)}(\mathbf{r})$ from Section 3.2:

$$
\hat{\mathbf{C}}_i(\mathbf{r}) = (1 - \alpha^{(\tau)}(\mathbf{r}))\hat{\mathbf{C}}_i^{(a)}(\mathbf{r}) + \alpha^{(\tau)}(\mathbf{r})\hat{\mathbf{C}}_i^{(\tau)}(\mathbf{r}).
\tag{11}
$$

The RGB loss for ray $\mathbf{r}$ with given color $\mathbf{C}(\mathbf{r})$ is:

$$
\mathcal{L}_{\text{rgb}}(\mathbf{r}) = \frac{\|\hat{\mathbf{C}}_i(\mathbf{r}) - \mathbf{C}_i(\mathbf{r})\|_2^2}{2\beta_i^{(\tau)}(\mathbf{r})^2} + \frac{\log \beta_i^{(\tau)}(\mathbf{r})^2}{2} + \lambda_\alpha \alpha^{(\tau)}(\mathbf{r}),
\tag{12}
$$

where $\beta_i^{(\tau)}(\mathbf{r}) = \alpha^{(\tau)}(\mathbf{r})\tilde{\beta}_i^{(\tau)}(\mathbf{r}) + \beta_{\min}$. As in NeRF-W, the uncertainty $\beta_i^{(\tau)}(\mathbf{r})$ is modeled as the variance of an isotropic normal distribution with mean $\hat{\mathbf{C}}_i(\mathbf{r})$, resulting the negative log-likelihood loss of $\mathbf{C}_i(\mathbf{r})$. It reduces the impact of transient occluders as noise when training static objects. $\beta_{\min} > 0$ is a hyperparameter for ensuring a minimum importance. The last term suppresses $\alpha^{(\tau)}$ as possible to prevent *TransientNet* from rendering static objects with $\lambda_\alpha > 0$. We set $\beta_{\min}$ to 0.1 and $\lambda_\alpha$ to 1.0.

## 3.4 Transient-aware depth prior

This section is about incorporating geometric prior from monocular depth into NeRF, which is highly inspired by NoPe-NeRF [24]. The mono-depth prior provides strong geometry cues for jointly training pose and NeRF. Even though NoPe-NeRF proposes inter-frame loss to leverage a depth prior, it can only be applied to the successive images from a video so we cannot adopt such loss for the unconstrained images. So, we apply only the depth loss Eq. (13). The predicted inverse depth $\{\mathbf{D}_i^{\text{inv}}\}_{i=1}^N$ is acquired from depth network DPT [50]. Since this is a monocular prediction, it is not an absolute inverse depth value. Therefore learnable parameters $\{\eta_i\}_{i=1}^N$ and $\{\xi_i\}_{i=1}^N$, which denote a scale and shift factor, were introduced for each mono-depth to learn undistorted depth map $\mathbf{D}_i^*$. It is optimized with rendered NeRF depth $\hat{\mathbf{D}}_i$:

$$
\mathbf{D}_i^*(\mathbf{r}) = \frac{1}{\eta_i \mathbf{D}_i^{\text{inv}}(\mathbf{r}) + \xi_i}, \quad \hat{\mathbf{D}}_i(\mathbf{r}) = \sum_{k=1}^K T(t_k)\alpha(\sigma(t_k)\delta_k)t_k.
\tag{13}
$$

By optimizing the L1 loss $\mathcal{L}_{\text{depth}}(\mathbf{r}) = \|\mathbf{D}_i^*(\mathbf{r}) - \hat{\mathbf{D}}_i(\mathbf{r})\|$, NeRF learns a strong geometry prior. However, since NeRF should exclude transient objects, we need to apply depth prior only to the areas without transient objects. In Section 3.1, the uncertain parts are generated from the candidate density $\sigma_i^{(c)}$ of the candidate head, while certain parts are generated from the shared density $\sigma$. This implies that the portion of candidate density also filters the transient parts as in Fig. 5. Based on this observation, we can measure transient confidence weight $\mathcal{W}_i^{\text{depth}}(\mathbf{r}) \in [0, 1]$ from the ratio of candidate density $\sigma_i^{(c)}$ to shared density $\sigma$. In consequence, the depth loss is defined as follows:

$$
\begin{aligned}
\mathcal{L}_{\text{depth}}(\mathbf{r}) &= \big(1 - \mathcal{W}_i^{\text{depth}}(\mathbf{r})\big)\|\mathbf{D}_i^*(\mathbf{r}) - \hat{\mathbf{D}}_i(\mathbf{r})\|, \\
\text{where } \mathcal{W}_i^{\text{depth}}(\mathbf{r}) &= \sum_{k=1}^K T_i^{(c)}(t_k)\alpha(\sigma_i^{(c)}(t_k)\delta_k)).
\end{aligned}
\tag{14}
$$

Table 1: Novel view synthesis results on PhotoTourism Dataset.

| | PSNR ↑ | | | | | SSIM ↑ | | | | | LPIPS ↓ | | | | |
| --- | --- | --- | --- | --- | --- | --- | --- | --- | --- | --- | --- | --- | --- | --- | --- |
| | ref. NeRF-W | BARF | BARF -W | BARF -WD | UP- NeRF | ref. NeRF-W | BARF | BARF -W | BARF -WD | UP- NeRF | ref. NeRF-W | BARF | BARF -W | BARF -WD | UP- NeRF |
| Brandenburg Gate | 26.50 | 8.94 | 11.98 | 12.45 | **26.29** | 0.887 | 0.221 | 0.629 | 0.619 | **0.884** | 0.116 | 1.027 | 0.549 | 0.630 | **0.126** |
| Trevi Fountain | 22.26 | 11.59 | 14.34 | 16.10 | **22.17** | 0.706 | 0.152 | 0.442 | 0.576 | **0.691** | 0.211 | 1.096 | 0.746 | 0.487 | **0.235** |
| Taj Mahal | 25.15 | 7.73 | 14.13 | 12.90 | **24.87** | 0.858 | 0.124 | 0.633 | 0.624 | **0.834** | 0.165 | 1.053 | 0.677 | 0.659 | **0.209** |
| Sacre Coeur | 22.33 | 8.25 | 12.51 | 9.33 | **21.59** | 0.827 | 0.229 | 0.679 | 0.651 | **0.791** | 0.123 | 1.027 | 0.571 | 0.453 | **0.157** |
| Mean | 24.06 | 9.13 | 13.24 | 12.70 | **23.73** | 0.820 | 0.182 | 0.596 | 0.618 | **0.800** | 0.154 | 1.051 | 0.636 | 0.557 | **0.182** |

Table 2: Camera pose estimation on PhotoTourism Dataset.

| | Rotation (°) ↓ | | | | Translation ↓ | | | |
| --- | --- | --- | --- | --- | --- | --- | --- | --- |
| | BARF | BARF-W | BARF-WD | UP-NeRF | BARF | BARF-W | BARF-WD | UP-NeRF |
| Brandenburg Gate | 128.32 | 21.42 | 159.00 | **0.621** | 4.479 | 2.890 | 3.334 | **0.070** |
| Trevi Fountain | 114.63 | 33.09 | 23.08 | **1.590** | 6.627 | 4.763 | 4.292 | **0.105** |
| Taj Mahal | 94.06 | 143.30 | 51.28 | **0.549** | 6.054 | 5.252 | 3.875 | **0.120** |
| Sacre Coeur | 99.33 | 112.19 | 113.50 | **1.452** | 12.353 | 9.875 | 9.073 | **0.222** |
| Mean | 109.09 | 77.50 | 86.72 | **1.053** | 7.378 | 5.695 | 5.144 | **0.129** |

## 3.5 Training pipeline

The overall training pipeline is shown in Fig. 2. Integrating the depth loss into Eq. (6) modified from Section 3.2 and Section 3.3, the overall loss is as follows:

$$\mathcal{L} = w_{u,v}\mathcal{L}_{\text{rgb}} + (1 - w_{u,v})(\mathcal{L}_{\text{feat}} + \lambda_d \mathcal{L}_{\text{depth}}). \tag{15}$$

The loss implies that depth loss $\mathcal{L}_{\text{depth}}$ decays as the training progresses as feature loss $\mathcal{L}_{\text{feat}}$ does. This is because we exploit depth supervision only for initial pose estimation, and using it further can degenerate the rendering quality in that the geometry prior we utilize is not a ground truth but like a pseudo-label from a depth network. A hyperparameter $\lambda_d$ for depth loss is set to 0.001.

We use a hierarchical sampling strategy, and there is a slight difference in the coarse model loss with Eq. (12). We optimize the TransientNet with the fine model alone, and the coarse model uses detached $\alpha^{(\tau)}$ and $\hat{\mathbf{C}}_i^{(\tau)}(\mathbf{r})$ to render $\hat{\mathbf{C}}_i(\mathbf{r})$ in Eq. (11), and does not use uncertainty. Thus, the final RGB loss for the coarse model is as follows.

$$\mathcal{L}_{\text{rgb}}^{\text{coarse}} = \frac{1}{2}\|\hat{\mathbf{C}}_i(\mathbf{r}) - \mathbf{C}_i(\mathbf{r})\|_2^2. \tag{16}$$

In addition, the way of composing a sample set for the fine model is slightly different from NeRF [1], which only uses the density $\sigma$ of the coarse model. Along with $\sigma$, we also need to use candidate density $\sigma^{(c)}$ until scheduling eliminates the effect of candidate head. Depending on the scheduling factor $w_{u,v}$, both density $\sigma$ and $\sigma^{(c)}$ are used at the beginning, and only $\sigma$ is used in the end.

## 4 Experiments

**Datasets.** To demonstrate our methods work well in the unconstrained images. We report results on the Phototourism dataset. It consists of internet photo collections of famous landmarks and we select 4 scenes, *Brandenburg Gate*, *Sacre Coeur*, *Taj Mahal*, and *Trevi fountain*, which are also used in NeRF-W. We follow the split used by NeRF-W [18] and downsample each image by 2 times. All the initial camera poses are set to the identity transformation.

**Implementation details.** All the models are trained for 600K iterations with randomly sampled 2048 pixel rays at each step with a learning rate of $5 \times 10^{-4}$ decaying to $5 \times 10^{-5}$ for NeRF and transient network $\mathcal{T}$, and $2 \times 10^{-3}$ decaying to $1 \times 10^{-3}$ for pose $\mathbf{p}$ and two factors $\mathbf{s}_i$ and $\mathbf{t}_i$ of depth. We use Adam optimizer [51] across all the experiments except test-time appearance optimization, where AdamW [52] is used instead. The number of sampling points in each ray for volumetric rendering is set to 128 for both coarse and fine models. We use the default coarse-to-fine strategy

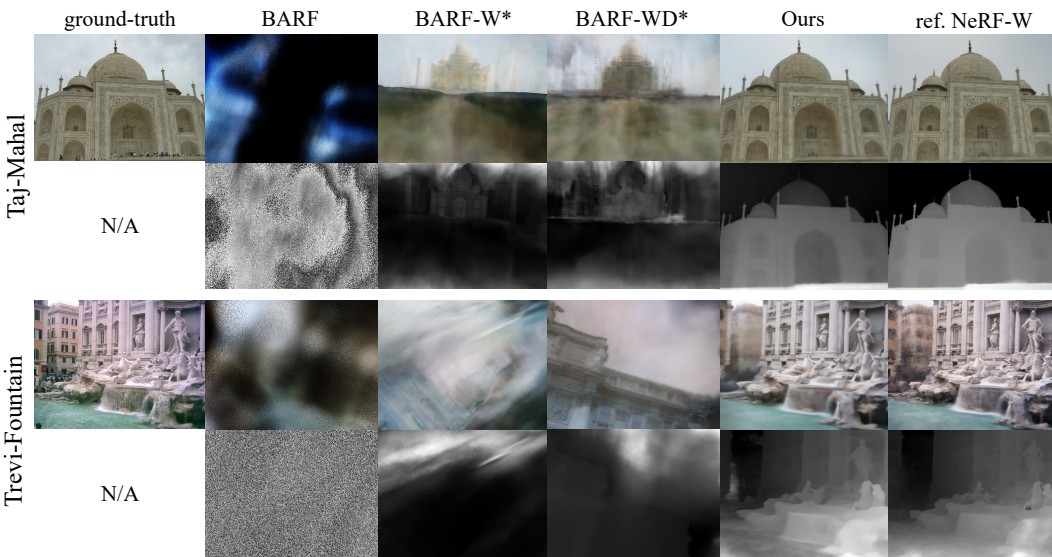

Figure 3: Qualitative results of baselines and our model, where (*) denotes BARF-variant baselines we implement. As shown in the figure (columns 2-4), a naive combination of BARF and other methods failed to converge, resulting in poor synthesis quality. In contrast, our model achieves comparable quality to reference NeRF with perfect camera poses.

of BARF which starts from training progress 0.1 to 0.5. We set the scheduling parameters $u$ and $v$ to be 0.1 and 0.5, respectively, the same as the parameters of coarse-to-fine. Note that both play the same role in helping robust pose estimation in the early stages. The transient confidence weight $\mathcal{W}^{\text{depth}}$ is manually detached to prevent those parameters from affecting NeRF directly. We extract deep features from the DINO model [49, 53], whose powerful 2D correspondence representation has been proven in various works [47, 54, 55], and monocular depth from DPT [50] offline. The detailed process of extracting features follows Amir et al. [49]. Full implementation details are provided in the supplementary material.

**Evaluation.** We implement the evaluation process which consists of two stages, test-time pose optimization, and appearance optimization. In NeRF-W, they do just appearance optimization since it trains with given poses. But in our case, we need to optimize pose either to evaluate novel view synthesis. Therefore, we optimize both pose and appearance on test images for pose test time optimization and then initialize appearance again and optimize it with the optimized pose. In conclusion, the results of novel view synthesis are evaluated using PSNR, SSIM [56], and LPIPS [57]. The train camera poses are Procrustes-aligned for comparison with ground truth poses in the same way as BARF [23].

### 4.1 Results

To verify the difficulty of our task, we compare our model with BARF [23] and its two variants. First, we implement BARF-W as a baseline by applying latent appearance modeling and the transient head of NeRF-W to BARF. The other baseline is relevant to NoPe-NeRF [24]. NoPe-NeRF succeeded in training NeRF with no pose prior in the outdoor scene. But it uses a sequence of images which is from video, so most of their principal components are using near-frame images. It is a strong prior to estimate camera poses, but in-the-wild images do not have such a prior so only the mono-depth loss is applicable to our setting. Therefore, we add the loss term Eq. (14) without $\mathcal{W}_i^{\text{depth}}$ to BARF-W (BARF-WD) to check whether the depth prior alone is enough to guide the model to robust pose estimation with unconstrained photos.

Table 2 shows the quantitative result of camera pose estimation, and translation errors are scaled by 10. As expected, BARF fails to estimate the poses due to the lack of photometric consistency in the unconstrained image collection. BARF-W also fails to optimize, which means that it is a hard problem that cannot be solved simply by adding appearance modeling and transient head. Although

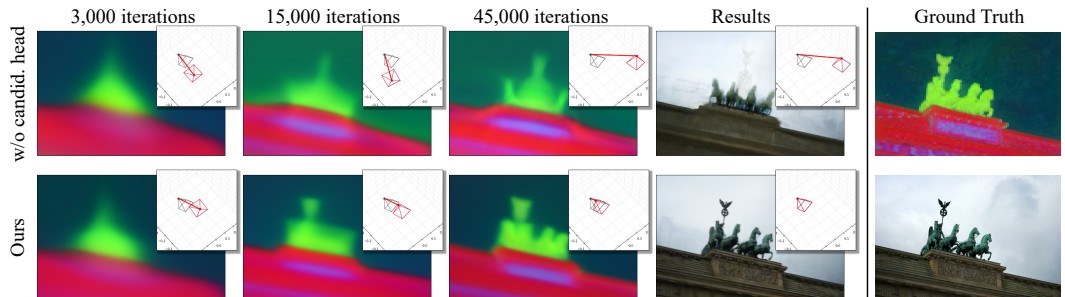

Figure 4: Visual comparison between our model with and without candidate head for a hard pose image during the training process. The small patch images compare the ground-truth pose (black) with the optimized poses (red).

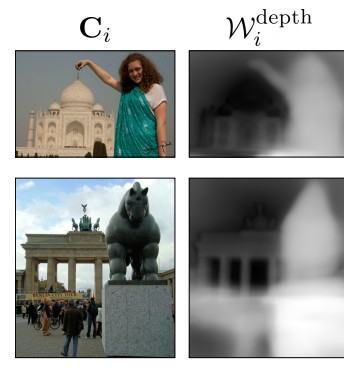

Figure 5: Visualization of transient confidence weight.

Table 3: Quantitative results of ablation study. All metrics are an average of 4 scene results.

| | Pose | | Novel View Synthesis | | |
| | Rot (°) ↓ | Trans↓ | PSNR ↑ | SSIM ↑ | LPIPS ↓ |
|---|---|---|---|---|---|
| UP-NeRF | **1.053** | **0.129** | **23.73** | **0.800** | **0.182** |
| w/o $\mathcal{L}_{\text{feat}}$ | 143.4 | 6.193 | 13.01 | 0.568 | 0.644 |
| w/o $\mathcal{T}_{\phi}$ | 2.329 | 0.314 | 22.51 | 0.781 | 0.224 |
| w/o $\mathcal{W}_i^{\text{depth}}$ | 1.226 | 0.170 | 23.60 | 0.795 | 0.215 |
| w/o $\mathcal{W}_i^{\text{depth}}, \ell_i^{(c)}$ | 4.972 | 1.042 | 21.23 | 0.744 | 0.283 |

BARF-WD showed slightly better results, it can be seen that the depth prior alone is not enough to solve the photometric inconsistency. In contrast, our model succeeds in pose estimation and shows a quality result.

Table 1 and Fig. 3 show the novel view synthesis results. As shown in the figure, baselines have poor rendering quality, which must be rooted in a failure of pose estimation. Even though variants of BARF (columns 3-4) show a slight improvement over BARF, its sub-optimal quality and pose estimation gives strong evidence that a mere combination of current methods or imposing depth supervision is not enough to resolve the problem. In contrast, our model shows comparable performance against reference NeRF-W and a clear appearance that implies its accurate pose estimation.

## 4.2 Ablation study

In this section, we verify the efficacy of each component in pose estimation with wild photos and provide a visualization of how these help the model optimize pose correctly.

**Feature-surrogate bundle adjustment.** In Section 3.2, we optimize the radiance field by minimizing $\mathcal{L}_{\text{feat}}$ for the early training since the unconstrained images are not guaranteed color consistency. When training only with photometric loss $\mathcal{L}_{\text{rgb}}$ without feature-surrogate, it can be seen from Table 3 that it fails to find accurate camera poses and render the novel views. This result shows that the feature-surrogate method is important for the early stage of training.

**Transient-free pose optimization.** We demonstrate the effect of separating the transient network from NeRF by comparing our model with a baseline where the transient network remains the same as NeRF-W, meaning that transient occluders are rendered from 3D space as static objects. Table 3 (column 2) clearly shows that gradient flows of transient objects adversely affect both pose estimation and NeRF optimization.

Table 4: Quantitative analysis on the candidate embedding size.

| size of $\ell_i^{(c)}$ | 0 | 8 | 16 | 32 |
|---|---|---|---|---|
| Rotation (°) ↓ | 4.972 | 1.286 | **1.053** | 3.162 |
| Translation ↓ | 1.042 | 0.184 | **0.129** | 0.474 |
| PSNR ↑ | 21.23 | 23.54 | **23.73** | 23.11 |
| SSIM ↑ | 0.744 | **0.802** | 0.800 | 0.779 |
| LPIPS ↓ | 0.283 | 0.184 | **0.182** | 0.201 |

Table 5: Overall learning time comparison with NeRF-W.

| | NeRF-W | | UP-NeRF | |
|---|---|---|---|---|
| | Preprocessing (COLMAP) | Training | Preprocessing (DINO&DPT) | Training |
| Brandenburg Gate | 29h 29m | 50h 34m | 0h 16m | 42h 02m |
| Trevi Fountain | 150h 06m | 51h 24m | 0h 36m | 42h 15m |
| Taj Mahal | 21h 33m | 49h 22m | 0h 20m | 42h 13m |
| Sacre Coeur | 25h 27m | 50h 34m | 0h 18m | 42h 51m |

**Transient-aware depth prior.** As shown in Fig. 5, confidence score filters transient occluders while imposing depth prior to the static correctly as we intended. Even though the static (*e.g.*, the gate) is slightly masked, it is justifiable in that the figure is visualized at the early of the training, meaning that the pose is not fully optimized and the model is still being trained to discern the static from the transient.

**Candidate head.** The last row of Table 3 shows that candidate head plays a crucial role in optimizing camera pose accurately. Additionally, we provide quantitative analysis on the candidate embedding $\ell_i^{(c)}$ size in Table 4. We observed that setting the candidate embedding to an appropriate size is of great help to camera pose estimation, and eventually succeeded in synthesizing novel view images.

### 4.3 Analysis

**Candidate head.** Fig. 4 shows that candidate embedding facilitates appropriate pose estimation for a hard pose image, inducing the model not to vainly struggle to overfit the scene without updating the pose. In contrast to the baseline without the candidate head which is stuck in a local minima, our model successfully estimates the pose. The figure implies that if the rendered scene at the early stage is similar to the ground truth with an incorrect pose, the baseline tries to fit the scene by optimizing NeRF, rather than pursuing accurate pose optimization. However, our model is more robust against being trapped into pose local minima for the hard pose images.

**Overall training time** We compared overall learning time between NeRF-W [18] and UP-NeRF in Table 5. Before training NeRF-W, it is necessary to run COLMAP to get camera poses. The time required for COLMAP depends on the number of images, and the execution time becomes non-negligible as the number increases. For example, *Trevi fountain* with about 3200 images, takes nearly 1 week to finish. However, UP-NeRF takes just 36 minutes to prepare training, which includes extracting DINO feature and DPT depth maps. The training time of UP-NeRF is also faster than NeRF-W because it renders transient objects directly from the feature maps without a volume rendering process. UP-NeRF only requires about 30 minutes of preprocessing time, so we expect using faster models such as Instant-NGP [58] to reduce the training time greatly.

## 5 Conclusion

We propose UP-NeRF, a robust unposed-NeRF that can learn with an unconstrained image collection with variable illumination and transient occluders. Thanks to the color-insensitive feature field, separated transient network, candidate head for robust pose optimization, and transient-aware depth prior, our model has less difficulty in estimating poses even in such challenging conditions. The experiment shows the promising results of our model compared to baselines, even comparable to the reference model with accurate poses.

## Acknowledgments and Disclosure of Funding

This work was partly supported by ICT Creative Consilience program (IITP-2023-2020-0-01819) supervised by the IITP, and the National Research Foundation of Korea (NRF) grant funded by the Korea government (MSIT) (NRF-2023R1A2C2005373).

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
