# UP-NeRF: Unconstrained Pose-Prior-Free Neural Radiance Fields (Supplement)

**Injae Kim**[*]
Korea University
dna9041@korea.ac.kr

**Minhyuk Choi**[*]
Korea University
sodlqnf123@korea.ac.kr

**Hyunwoo J. Kim**[†]
Korea University
hyunwoojkim@korea.ac.kr

In this supplementary material, we provide additional implementation details (Appendix A) of our model and visualization of ablation studies (Appendix B) which are not included in our main paper. Our experiments are based on PyTorch [1] with GeForce RTX 3090 GPU. To follow the original implementation as possible, we have BARF experiments on the original code[3] with a few modifications for compatibility with the Phototourism dataset, and implementation of NeRF-W, BARF-W, and BARF-WD are based on [2] because there is no official NeRF-W code available.

## A  Implementation details

The detailed architecture of UP-NeRF is shown in the Fig. 1. The size of each embedding is 48 for $\ell^{(a)}$, 128 for $\ell^{(\tau)}$, and 16 for $\ell^{(c)}$. At training, we sample 128 points uniformly with noise for a coarse model, between a predefined near and far parameter (0.1 and 5 each).

**Positional encoding.**    We use the same setting with BARF [3] for the positional encoding with L frequency bases (10 for 3D point $\mathbf{x}$ and 4 for viewing direction $\mathbf{d}$), defined as:

$$\gamma(\mathbf{x}) = [\mathbf{x}, \gamma_0(\mathbf{x}), \gamma_1(\mathbf{x}), \dots, \gamma_{L-1}(\mathbf{x})] \in \mathbb{R}^{3+6L},$$
$$\gamma_k(\mathbf{x}) = [\cos(2^k \pi \mathbf{x}), \sin(2^k \pi \mathbf{x})] \in \mathbb{R}^6. \tag{1}$$

We also use the coarse-to-fine strategy in BARF:

$$\gamma_k(\mathbf{x}; \rho) = w_k(\rho) \cdot [\cos(2^k \pi \mathbf{x}), \sin(2^k \pi \mathbf{x})] \in \mathbb{R}^6, \tag{2}$$

where the weight $w_k$ is defined as:

$$w_k(\rho) = \begin{cases} 0 & \text{if } \rho < k \\ \dfrac{1 - \cos((\rho - k)\pi)}{2} & \text{if } 0 \leq \rho - k < 1 \\ 1 & \text{if } \rho - k \geq 1. \end{cases} \tag{3}$$

We linearly increase the value $\rho$ from 0 to $L$ between 0.1 and 0.5 for the entire training iteration and finally activate all frequency bands ($L$).

**Feature encoder and mono-depth encoder.**    UP-NeRF uses two pretrained models, DINO [4, 5] for deep feature map $\mathbf{F}_i$ and DPT [6] for mono-depth map $D_i$. For the DINO feature encoder, we use the implementation[4] by Amir et al [5]. We use dino_vits8 checkpoint and get the feature at the 9th layer. This implementation obtains a higher-resolution feature map by overlapping patches with stride 4. All the images are resized 448 by 448 before encoding and the feature map of size of 111 by 111 is obtained. For a DPT mono-depth encoder, we use dpt_large-midas checkpoint of the official

---

[*]First two authors have an equal contribution.

[†]Corresponding author.

[3]https://github.com/chenhsuanlin/bundle-adjusting-NeRF

[4]https://github.com/facebookresearch/dino

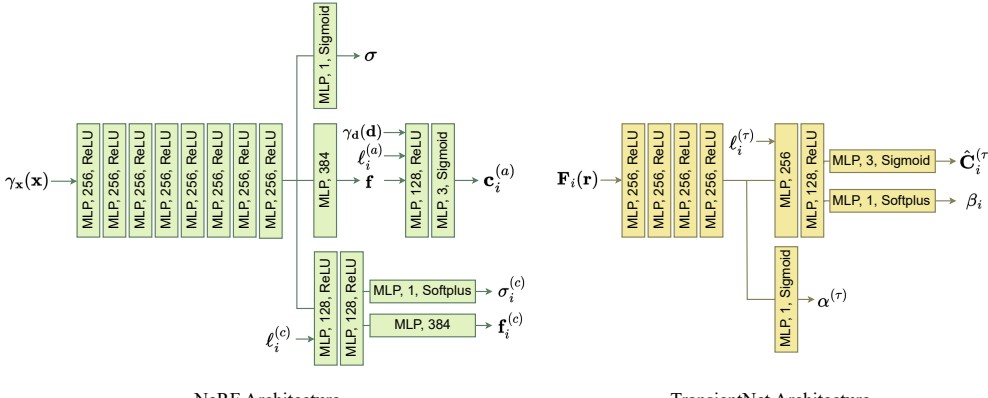

Figure 1: Detailed architecture of our method.

implementation[5]. The obtained mono-depth maps are normalized between 0.1 and 5 based on the predefined near and far parameters. The low-resolution feature map and depth map are resized by bilinear interpolation to recover the resolution.

**Other models.**  As we mentioned, NeRF-W [7] follows the implementation of [2]. BARF-W is implemented by replacing given camera poses with the learnable parameters in NeRF-W, and it optimizes those parameters with a coarse-to-fine strategy. BARF-WD is implemented by imposing additional mono-depth loss from NoPe-NeRF [8] to BARF-W. The depth to which mono-depth loss is imposed is rendered using both static and transient density, which follows the architecture of NeRF-W where the transient network is not isolated as opposed to ours.

For the ablation study of isolated TransientNet $\mathcal{T}_\phi$, we implement the model to predict transient color $\mathbf{c}_i^{(\tau)}$, transient sigma $\sigma_i^{(\tau)}$, and $\beta_i$ using predicted candidate feature $\mathbf{f}_i^{(c)}$ and candidate embedding $\ell_i^{(c)}$, since we observed that candidate head is capable of filtering transient objects as well as enabling robust pose optimization which is its original role. Transient embedding $\ell_i^{(\tau)}$ is not used in this case because the candidate head also functions as a transient head on top of its original role, so it is of no use separating two embeddings $\ell_i^{(c)}$ and $\ell_i^{(\tau)}$. Then we render the pixel color $\hat{\mathbf{C}}_i(r)$ using $\mathbf{c}_i$, $\mathbf{c}_i^{(\tau)}$, $\sigma_i$, and $\sigma_i^{(\tau)}$ as NeRF-W does.

**Test-time optimization.**  As we mentioned in the main paper, the evaluation process entails two stages, which are test-time pose optimization and appearance optimization. BARF has only pose optimization because BARF is not capable of considering different appearances, and NeRF-W has only appearance optimization because it trains with given poses. The rest of the models including ours have both optimizations. To optimize poses, we freeze all the parameters except the pose parameter and appearance embedding, initialize these two parameters, and optimize them with test images. We initialize pose parameters of test images as follows: First, using trained poses of train images and ground-truth poses of them, we can obtain the alignment matrix of these two groups of poses using Procrustes analysis. Then we apply this alignment to the ground-truth test poses and use it as an initial pose parameter for test-time optimization. Then, appearance embeddings are re-initialized with the pose parameters frozen, and optimized again with test images, but only with the left half of the images at this time. Evaluation is done only with the right half of the images as NeRF-W does to minimize information leakage. We report the best PSNR, SSIM, and LPIPS during evaluation. Some hyperparameters are different from the training. First, optimization is done with 20k iterations for pose optimization and 20 epochs for appearance optimization. Second, learning rates are $5 \times 10^{-3}$ for appearance embeddings and $1 \times 10^{-4}$ for pose parameters in pose optimization, and $1 \times 10^{-1}$ for appearance embeddings in appearance optimization without any decay. Lastly, the number of sampled rays is reduced from 2048 to 1024, and we sample 128 points for the coarse model and 128 points for the fine model on each ray.

---

[5]https://github.com/isl-org/DPT

Table 1: Novel view synthesis results on additional Phototourism scenes.

| | View Synthesis Quality | | | | | | | | | Camera Pose Estimation | | | |
| | PSNR ↑ | | | SSIM ↑ | | | LPIPS ↓ | | | Rotation (°) ↓ | | Translation ↓ | |
| | ref. NeRF-W | BARF | Ours | ref. NeRF-W | BARF | Ours | ref. NeRF-W | BARF | Ours | BARF | Ours | BARF | Ours |
|---|---|---|---|---|---|---|---|---|---|---|---|---|---|
| British Museum | 19.03 | 9.606 | **18.11** | 0.736 | 0.147 | **0.676** | 0.270 | 1.076 | **0.303** | 158.8 | **5.516** | 2.644 | **0.328** |
| Lincoln Memorial Statue | 22.95 | 9.111 | **21.89** | 0.719 | 0.170 | **0.760** | 0.323 | 0.994 | **0.347** | 173.2 | **2.750** | 6.439 | **1.340** |
| Pantheon Exterior | 22.24 | 9.742 | **21.10** | 0.823 | 0.144 | **0.814** | 0.127 | 1.050 | **0.143** | 163.3 | **1.454** | 6.035 | **0.230** |
| St.Pauls Cathedral | 22.75 | 8.226 | **22.69** | 0.784 | 0.239 | **0.783** | 0.154 | 0.984 | **0.192** | 92.4 | **2.531** | 6.072 | **0.498** |
| Mean | 21.74 | 9.171 | **20.95** | 0.766 | 0.175 | **0.758** | 0.219 | 1.026 | **0.246** | 147.0 | **3.063** | 5.298 | **0.599** |

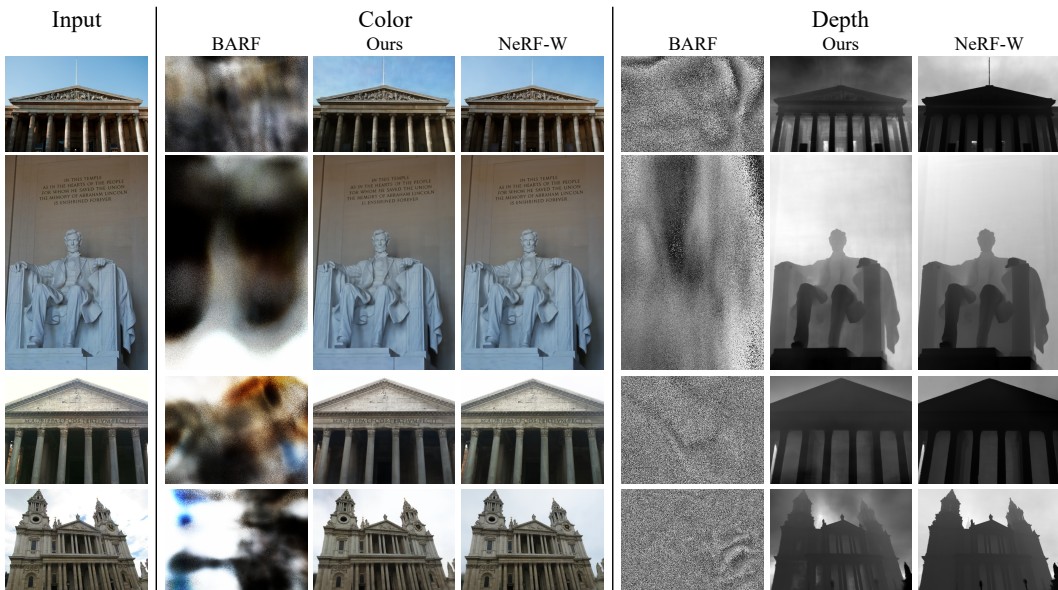

Figure 2: Qualitative visualization of additional Phototourism scenes.

# B  Additional experiments

**Ablation study of candidate head.**   In Fig. 3, we show the visualization of additional ablation study of the candidate head in other scenes. During the early stage of training, only the feature field is learned, so we show a feature map by using a 3-dimensional PCA component. It is shown that without using the candidate head, the model is more prone to fall into local minima during pose estimation, failing to learn the correct poses.

**Ablation study of TransientNet.**   Table 3 shows that not separating the transient head has a bad influence both on pose estimation and rendering quality. The failed cases of the model without separation are shown in Fig. 4. It can be observed that the model failed to estimate the correct poses, which suggests the necessity of such separation.

**Transient filtering.**   Fig. 5 shows the visual comparison of the decomposition of transient occluders between UP-NeRF and NeRF-W. Without pose prior, our model achieves comparable or even superior results to NeRF-W, demonstrating the effectiveness of transient filtering using deep features. Furthermore, the visualization of decomposed transient occluders (column 3) implies that our model succeeded in handling the cases where NeRF-W misjudged transient occluders as static objects.

**Additional scenes.**   We think presenting additional Phototourism scenes other than 4 scenes can highlight the robustness of our model in the wild scenes. Thus, we pick several other scenes (*British museum, Lincoln Memorial statue, Pantheon Exterior, St. Paul's Cathedral*) for additional experiments. We follow the preprocessing step of NeRF-W, which filters out images whose NIMA [9]

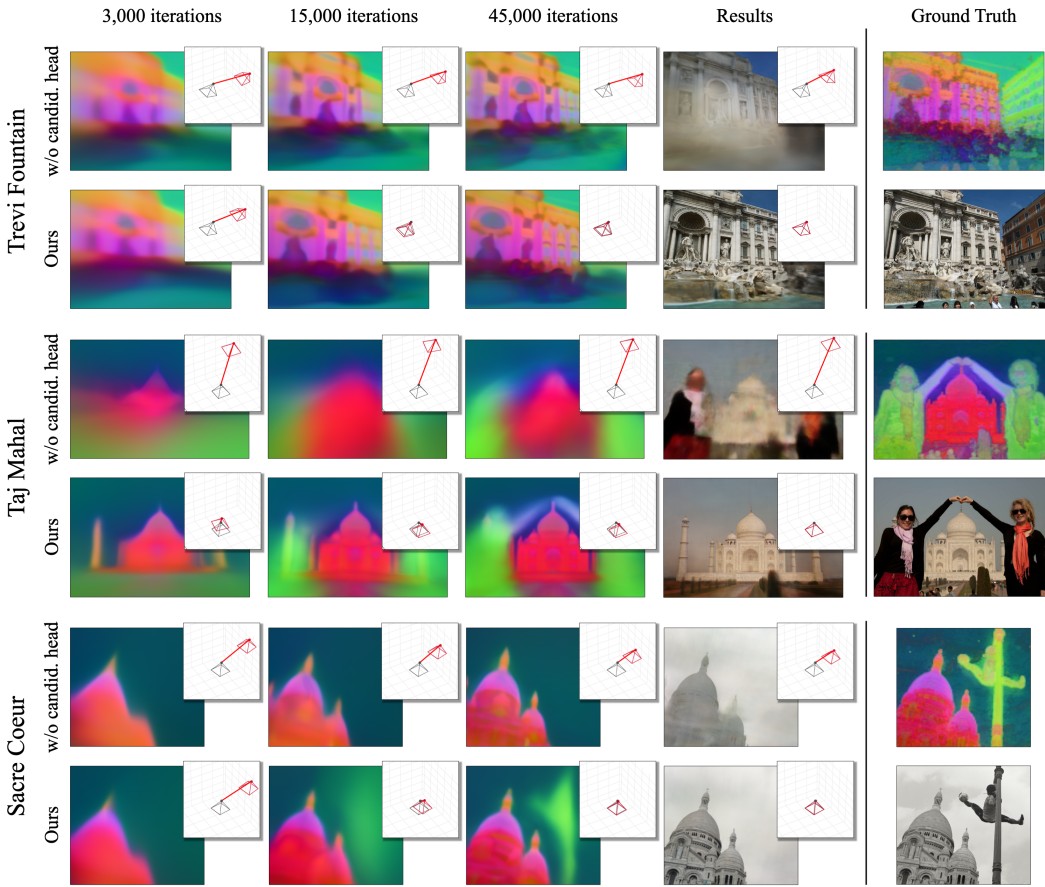

Figure 3: Additional visual comparison between our model with and without candidate head during the training process. The small patch images compare the ground-truth pose (black) with the optimized poses (red).

score is below 3, and where transient objects occupy more than 80% of the image using DeepLab v3 [10] model. As NeRF-W didn't clarify the process of selecting test images other than hand-selecting, we arbitrarily set criteria of being candidate of test images: NIMA score must be in the 70th percentile, and occupancy of transient occluders must be in the bottom 25%. Then, we pick test images manually with the least transient occluders as possible. Qualitative results and quantitative comparisons with BARF and NeRF-W are provided in Table 1 and Fig. 2.

## C Limitations

The performance of our method is influenced by pretrained models, DINO and DPT. If the pretrained models have low performance, our model is likely to have lower performance as well. Though it optimizes the radiance field without using the feature map and mono-depth map in the later stage of training, the problem still remains in that TransientNet utilizes features as input.

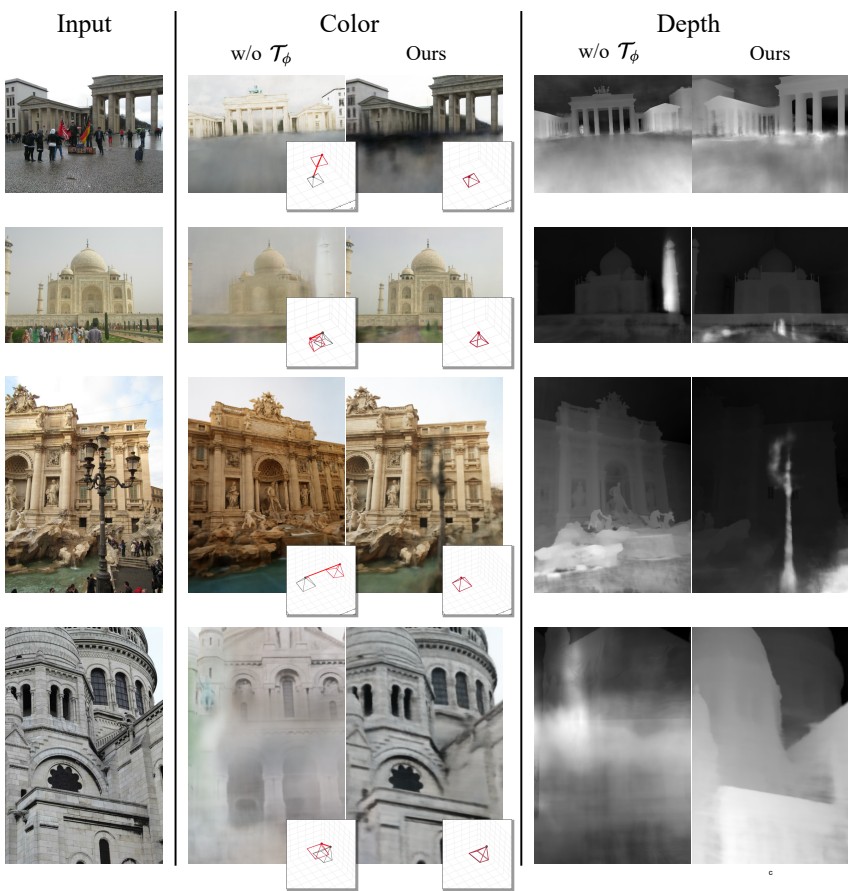

Figure 4: Visual comparison of ablation study of the isolated transient network. It is shown that our model succeeds in estimating the correct pose as opposed to the baseline which fails to estimate the correct poses and eventually renders the scene from different viewpoints. The small patch images compare the ground-truth pose (black) with the optimized poses (red).

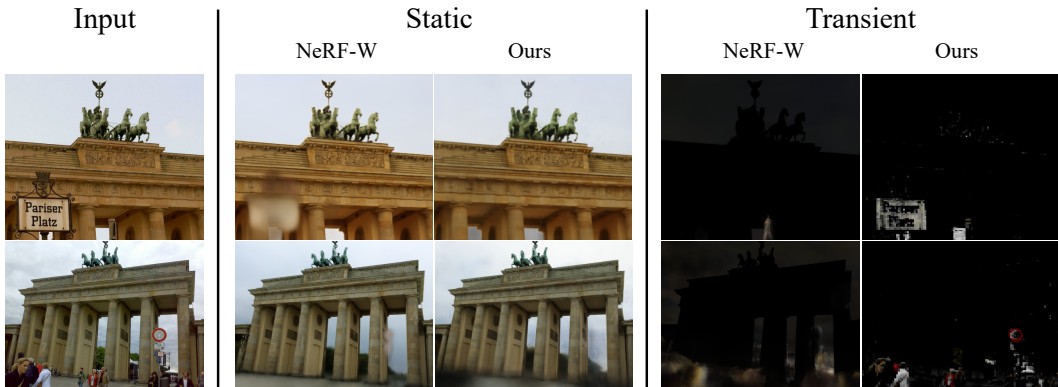

Figure 5: Visual comparison of decomposition of transient objects between our model with reference NeRF-W. Our model achieves comparable or even superior decomposition to NeRF-W even though our model is given no pose prior unlike NeRF-W. The transient figure of ours (column 5) is visualization after alpha composition.