# OpenReview forum: "UP-NeRF: Unconstrained Pose Prior-Free Neural Radiance Field"
_NeurIPS.cc/2023/Conference — NeurIPS 2023 poster_

### Official Review · Reviewer_FYmx · 2023-07-02

**Soundness:** 3 good
**Presentation:** 2 fair
**Contribution:** 2 fair
**Rating:** 6
**Confidence:** 5

**Summary:**

This paper solves an interesting problem — joint pose and NeRF optimization on in-the-wild image collections. Unlike prior works such as BARF, this work aims at handling unconstrained images with varying illumination and transient occluders. To tackle this problem, this work incorporates learnable camera parameters, depth prior, and semantic features from DINO as supervision into the NeRF-W framework. Experiments demonstrate that the proposed method outperforms BARF and its variants on unstructured Internet images.

**Strengths:**

1.  This work identifies an interesting and important research problem — joint pose and NeRF optimization on in-the-wild images. Prior works on the joint pose and NeRF optimization are limited to controlled settings.
2. The proposed method is intuitive and combines the NeRF-W framework and several components (e.g., depth prior, semantic features, etc.) from other works (such as NoPe-NeRF).

**Weaknesses:**

1. The main contribution of this paper is the problem setting — pose and NeRF estimation in the wild. The proposed method combines NeRF-W and BARF, which has limited technical novelty.
2. The presentation may be improved. Figure 1 and Figure 2 have not been referred to and explained in the main text.
3. Missing references:  Meng et al. GNeRF: GAN-based Neural Radiance Field without Posed Camera. ICCV 2021

**Questions:**

What is the effect of using semantic features as supervision, i.e. L_feat?


**Limitations:**

I did not find discussions on the limitations of this paper. It would be great to discuss the limitations and analyze the possible reasons and future works.

---

> ### Author Rebuttal · Authors · 2023-08-07
>
> **We appreciate your constructive feedback on our work and we will reflect them. Below are our responses to the reviewer's questions.**
>
> ---
> **Comment 1**: The proposed method has limited technical novelty.\
> **Answer**: As the reviewer admitted, Unposed-NeRF in the outdoor scene is a challenging problem, and only a handful of recent works (e.g., NoPe-NeRF) partially addressed this (clean images NOT in-the-wild setting). As reviewer qVYM and others mentioned, we believe that our work is novel. Furthermore, as shown by BARF-W in Table 1 and 2 , a simple combination of NeRF-W and BARF fails to optimize unposed-NeRF with unconstrained images.
> To solve the problem, we proposed 4 novel techniques: candidate head (Sec3.1), feature-surrogate bundle adjustment (Sec3.2), isolated transient network (Sec3.3) and transient-aware depth prior (Sec3.4).
>
> ---
> **Comment 2**: Figure 1 and Figure 2 have not been referred to and explained in the main text.\
> **Answer**: We will add more explanations in the main text with explicit figure numbers.
>
> ---
> **Comment 3**: Missing references.\
> **Answer**: Thank you for letting us know the missing reference. We will include these in the final version.
>
> ---
> **Comment 4**: What is the effect of using semantic features as supervision, i.e. L_feat? \
> **Answer**: As we mentioned in lines 149-152 of the main paper, because of the diverse appearances of images it is difficult to optimize the model with the photometric loss. In contrast, DINO features, or semantic features, are less sensitive to the variation of appearance, as shown in Figure 1. Therefore, by using semantic features as supervision, UP-NeRF succeeded in jointly optimizing NeRF and camera poses, unlike the prior methods that are based on photometric loss (See Table 1, 2 and Figure 3).
>
> ---
> **Comment 5**: I did not find discussions on the limitations of this paper.\
> **Answer**: We discussed the limitations in the supplement, and additionally shared some failure cases in the Fig. 3 in pdf.

---

> > ### Comment · Reviewer_FYmx · 2023-08-12
> > **Follow-up**
> >
> >
> > 1. For the response to "What is the effect of using semantic features as supervision, i.e. L_feat?"
> >
> > I understand the intuition behind this loss function. The proposed method involves many other objective functions besides the photometric loss commonly used in prior works. I would suggest adding a quantitative evaluation of the L_feat to Table 3.
> >
> > 2. For the response to "I did not find discussions on the limitations of this paper."
> >
> > I'm not sure if I understand "... additionally shared some failure cases in Fig. 3". The visualizations in Fig. 3 seem to be the cases where the proposed method "achieves comparable quality to reference NeRF with perfect camera poses".
> >
> > Nevertheless, this work explores a very interesting problem and provides a reasonable solution. I'm happy to raise my rating.

---

> > > ### Author Response · Authors · 2023-08-15
> > >
> > > We appreciate your support for our work.
> > >
> > > We provide the ablation study of $L_{\text{feat}}$ at Brandenburg Gate below, and will add other scene results and reflect them in the final version.
> > >
> > > | Method | w/o feature-surrogate | Ours |
> > > | --- | --- | --- |
> > > | Rotation (°) ↓ | ​　　　　　175.3 |​0.797 |
> > > | Translation ↓ | ​　　　　　3.846 | ​0.148 |
> > > | PSNR ↑ | ​　　　　　12.29 | 23.60 |
> > > | SSIM ↑ | ​　　　　　0.645 | ​0.801 |
> > > | LPIPS ↓ | ​　　　　　0.666 | 0.180 |
> > >
> > > ---
> > >
> > > We apologize for the inaccurate reference to “… in the Fig. 3 in pdf”. We added **additional PDF in Author Rebuttal** and some failure cases are in the Fig. 3 of this PDF, **not in the Fig.3 of main paper**. And you can find discussion of the limitations in **section C of supplementary PDF**.

---

### Official Review · Reviewer_Kmtq · 2023-07-02

**Soundness:** 3 good
**Presentation:** 2 fair
**Contribution:** 2 fair
**Rating:** 6
**Confidence:** 4

**Summary:**

This paper tackles NeRF training from in-the-wild Internet photos without pre-computed camera poses. The main idea is to leverage (self-supervised) image features and a carefully designed optimization strategy, together with ideas from existing work, including modeling transient regions and using mono-depth supervision. These components work in combination to avoid local minima and enable joint optimization of both pose and NeRF from scratch.

Experiments show that this joint optimization pipeline successfully recovers camera poses in several in-the-wild scenes and leads to good NVS results comparable to those with COLMAP preprocessing, whereas the previous work of BARF and its variants all fail.

**Strengths:**

### S1 - Technically sound approach to a challenging problem
- The task of estimating camera poses on raw in-the-wild Internet photos using a gradient-descent-based optimization is challenging, due to noisy correspondences and local minima.
- The proposed method incorporates many good ideas from existing work and works well on a number challenging scenes.
-- For instance, the use of self-supervised image features for registering semantic correspondences via rendering;
-- The mechanism to model transient objects using learned opacity. This paper further adapts the original volumetric opacity in NeRF-W to per-image opacity, which is claimed to be more effective.
-- The use of depth supervision from pre-trained models.

### S2 - Promising results
- The paper demonstrates good results on a number of challenging scenes using Internet photos, where existing methods clearly fails.

**Weaknesses:**

### W1 - Complicated pipeline
- The resulting pipeline is complicated, involving many components, eg, 6 MLPs in total, and easily becomes confusing. It took me several passes back and forth to understand the exact implementation.
- It also requires a heavily crafted training schedule, gradually activating and deactivating some of the components.
- A critical concern on such a pipeline is its robustness across various scenes. Would one need to fight against all the hyperparameters and the training schedule when training on other scenes. I strongly suggest the authors also present results on other standard datasets (W3).

### W2 - Unclear motivation and potential redundancy of some technical designs
- It is unclear to me why the per-image "candidate embeddings" can help with the pose optimization. It seems the motivation is to allow some of the per-image variations (multi-view inconsistencies) to be factored into these per-image embeddings. The paper presents ablation results which shows numerical benefits of this component, but why is it useful for pose estimation?
- Are they still useful in general in the case of a multi-view static scene without transient entities?
- Isn't this the job of the separate mechanism for handling transient regions?
- Also, it is confusing to me why the transient confidence weights $\mathcal{W}_i^{\text{depth}}$ are calculated from the candidate densities $\sigma^{(c)}$, rather than from the transient opacities $\alpha^{(\tau)}$. Are there some redundancy between the two?

### W3 - Only on one dataset
- The paper only presents results on one dataset, consisting of Internet photo collections of 4 scenes, which the method is specifically tailored to.
- However, the paper claims to solve general "unposed NeRF". I strongly recommend the authors to also test it on standard multi-view datasets, eg DTU, CO3D etc, to assess the robustness of this complicated pipeline.

### (minor) W4 - Do intrinsics matter?
- How are the camera intrinsics obtained? From COLMAP? If so, this would undermine the value of avoiding cumbersome COLMAP preprocessing.
- Or are they estimated or assumed to be some value for all images?

### Other comments
- There are a few existing work that leverages DINO for pose estimation of objects, which the authors should consider referencing, eg: Zero-Shot Category-Level Object Pose Estimation [1], LASSIE [2], MagicPony [3].
- Line 125: when the "candidate embeddings" are first introduced, it was not immediately obvious to me they are per-image embedding vectors. Also, are they jointly optimized? The term "candidate head" is also quite obscure to me.
- Fig 2: the blue arrow in the middle connecting $\hat{\mathbf{F}}_i^{(c)}$ to $\theta_4$ is slightly inaccurate, as the $\hat{\mathbf{F}}_i^{(c)}$ is the result of ray integration, whereas the input to $\theta_4$ should be the raw per-point feature $\hat{\mathbf{f}}_i$, if I understand correctly.
- What is the computational overheads compared to vanilla NeRF?

### References:
- [1] Zero-Shot Category-Level Object Pose Estimation. ECCV 2022.
- [2] LASSIE: Learning Articulated Shape from Sparse Image Ensemble via 3D Part Discovery. NeurIPS 2022.
- [3] MagicPony: Learning Articulated 3D Animals in the Wild. CVPR 2023.

**Questions:**

1. Clarify the motivation for the "candidate embeddings";
2. If possible, present more evaluation results on other datasets **with the same hyperparameters and training schedule**, to validate the robustness;
3. It might be difficult as the pipeline is indeed complicated, but if possible, it would be helpful to further simplify the notations, in particular, the super- and sub-scriptions.

**Limitations:**

The authors included a brief paragraph on the limitation of the image features. I would expect some discussion on the robustness of the proposed pipeline.

---

> ### Author Rebuttal · Authors · 2023-08-07
>
> **We appreciate the detailed comments and suggestions. We address all the questions below. We hope that our answers resolve the reviewer's concerns and lead to support.**
>
> ---
> **Answer of "W1 - Complicated pipeline with hyperparameters"**
>
> Although our method has several hyperparameters, the method is relatively robust to the choice of hyperparameters. We here analyze the effect of the candidate embedding size $\ell^{(c)}_i$. Our experiments on the phototourism dataset show that when the candidate embedding size is reasonably big $(\ge 16)$, it stably achieves good performance. All other hyperparameters are set in the same way as previous works.
>
> |Size of $\ell^{(c)}_i$ |0|16|32|64
> | --- | --- | --- | --- | --- |
> |Rotation (°) ↓|4.358|1.225|1.057|**1.047**
> |Translation ↓|0.843|0.164|**0.147**|0.152
> |PSNR ↑|21.45|23.26|**23.47**|23.40
> |SSIM ↑|0.745|**0.800**|0.799|**0.800**
> |LPIPS ↓|0.294|**0.181**|0.183|0.183
> ---
> **Answer of "W2-1 Motivation and more explanations of candidate head."**
> + **Candidate head (motivation and intuition)**: \
> Thank you for asking for more explanations of our contributions. We provided more explanations of the candidate head in the General Response (Author Rebuttal) above. Please refer to it.
> + **The usefulness of candidate embedding in the case of a multi-view static scene without transient entities**:\
> Our candidate head prevents the hard-pose images from falling into local minima. Handling transient regions is only an additional role. To validate the usefulness of candidate embedding, we had additional experiments on the BLEFF dataset presented by the authors of NeRF-- [3], which is a synthetic dataset without transient entities with several perturbations by random rotations and translations. As the table below shows, candidate embedding shows more robust results as the poses of images become more perturbed.
>
> |  Scene | Perturb | ​PSNR ↑  | | SSIM ↑ | | LPIPS ↓  | | Rot (°) ↓ | | Trans↓  | |
> | --- | --- | --- | --- | --- | --- | --- | --- | --- | --- | ----- | --- |
> |  |  |BARF|BARF+$\ell^{(c)}_i$|BARF |BARF+$\ell^{(c)}_i$|BARF   |BARF+$\ell^{(c)}_i$|BARF  |BARF+$\ell^{(c)}_i$|BARF  |BARF+$\ell^{(c)}_i$
> | Classroom | 20 | 25.43|26.13 (**+0.70**) | 0.877|0.893 (**+0.016**) | 0.133|0.105 (**-0.028**) | 0.265|0.293 (+0.028) | 0.009|0.022 (+0.013) |
> | Classroom |30 | 12.36|21.36 (**+9.00**) | 0.480|0.806 (**+0.326**) | 0.749|0.183 (**-0.566**) | 29.17|6.182 (**-22.99**) | 2.176|0.354 (**-1.822**) |
> ||||||||||||
> | Bed |20 | 37.42|37.06 (-0.36) | 0.969|0.967 (-0.002) | 0.050|0.048 (**-0.002**) | 0.839|0.296 (**-0.543**) | 0.243|0.230 (**-0.013**) |
> | Bed |30 | 13.61|32.47 (**+18.86**) | 0.449|0.890 (**+0.441**) | 0.897|0.142 (**-0.755**) | 141.8|7.077 (**-134.7**) | 16.57|10.43 (**-6.14**) |
>
> **Answer to "W2-2 Redundancy of some technical designs."**
> + **Redundancy between the transient opacities and the candidate densities**: \
> If we use the transient network in the early stage of training, it will generate static regions as well as transient regions because the camera poses are not learned. So we should use the candidate densities for transient confidence weights.
>
> ---
> **Answer of "W3 - Only on one dataset"**
>
> As mentioned, we had additional experiments on BLEFF dataset which is a standard multi-view dataset. We use the same hyperparameters and training schedule. We will add more scene results in the final version.
>
> ---
> **Answer of "W4 - (minor) W4 - Do intrinsics matter?"**:
>
> As previous works such as BARF, SPARF, and NoPe-NeRF, we assume that intrinsic parameters are given. Our work focuses only on estimating extrinsic parameters (camera pose). In real-world applications, we may get intrinsic parameters from image metadata. Also, we may leverage previous works [1, 2] to estimate the intrinsic parameters.
>
> ---
> **Other comment 1**: Missing references.
>
> **Answer**: Thank you for letting us know missing reference. We will include these in the final version.
>
> ---
> **Other comment 2**: Fig 2 is slightly inaccurate.
>
> **Answer**: We appreciate it for pointing out an inaccurate connection in our figure. The update is available in Fig. 1 in PDF.
>
> ---
> **Other comment 3**: What is the computational overheads compared to vanilla NeRF?\
> **Answer**: Vanilla NeRF is not suitable for in the wild dataset, so we conducted comparisons with NeRF-W. You can refer to the table below. NeRF-W needs to obtain camera poses with COLMAP, and it takes over 25 hours. However, 10 minutes is enough for UP-NeRF to prepare for training; the process of obtaining DINO features and DPT mono-depth predictions is included.
>
> |  | NeRF-W | | UP-NeRF | |
> | --- | --- | --- | :---: | --- |
> |  | Preprocessing (COLMAP) | Training | Preprocessing (DINO&DPT) | Training |
> | Brandenburg Gate | 29h 29m | 50h 30m | 10m | 46h 30m|
> | Sacre Coeur | 25h 27m | 50h 30m | 10m | 46h 30m|
>
> ---
> **Other comment 4**: Complicated notations.\
> **Answer**: We agree with your opinion, we will try to find out the way to simplify the notations.
>
> ---
> ### References:
> [1] DeepCalib: A Deep Learning Approach for Automatic Intrinsic Calibration of Wide Field-of-View Cameras
>
> [2] DeepPTZ: Deep Self-Calibration for PTZ Cameras
>
> [3] NeRF−−: Neural radiance fields without known camera parameters.

---

> > ### Comment · Reviewer_Kmtq · 2023-08-16
> > **Thanks for the responses**
> >
> > The rebuttal addressed my two main concerns pretty well. The additional explanation on the 'candidate head' makes the motivation much clearer. Also the additional evaluation on BLLFF and the ablation with the 'candidate head' on those static scenes are quite useful in demonstrating both the robustness and efficacy of the proposed approach. I would highly recommend including more results in the final version. I'm happy to raise my rating to accept.

---

> > > ### Author Response · Authors · 2023-08-21
> > >
> > > We appreciate your generous appraisal. We will include more comprehensive experiments and results in the final version.
> > > However, we observed that the rating remains unchanged. If it is not a mistake, we would be very grateful if you let us know the reason for this decision for further improvements of our work.

---

### Official Review · Reviewer_qVYM · 2023-07-04

**Soundness:** 3 good
**Presentation:** 2 fair
**Contribution:** 3 good
**Rating:** 7
**Confidence:** 4

**Summary:**

The paper produces a novel approach for optimisation of pose in NeRF scenarios. The core novelty is the addition of a candidate head that improves network stability when the images poses are not yet converged, along with some other tweaks like a transiency inference head or a feature field.

Edit: I have read the rebuttal, and given the scores from other reviewers and I would like to keep my score.

**Strengths:**

* The idea is novel, interesting and timely.
* The results show good improvements over the current s-o-t-a.
* The experiments section is thorough enough to demonstrate the qualities of the proposed approach.

Overall, while my negatives might sound long and would seen to outweigh the positives, I think it brings a potentially valuable contribution to the community, so I very much support the acceptance of the paper. My main worry is about the somewhat confusing and short explanation of the core contribution of the “candidate” heads, which should be expanded and improved.

**Weaknesses:**

* The paper is somewhat confusing to read at times. For example “unconstrained images” are introduced and used without introducing the term. The meaning does become later in the paper, but it would be good to introduce things earlier on. Similarly, I do not see much point for Figure 1, as features have been used for this type of matching for years.
* Even though the literature review section of the paper is quite good, I think the paper does anchor things a bit too much on BARF, and ignores other works such as NoPe-NERF / GARF / etc in the comparisons. I do understand that some of these works were arxiv at the time of the submission (but are publications now), so this should not penalise the paper too much. That being said, it’d be nice if some of these comparisons could be added.
* I also found some of the notation difficult to follow initially as I found not find any outline of it’s meaning. An example is the (c) in line 126 which is not explained.
* Looking at the core contribution of the paper (i) the intuition (i.e. the “roughly speaking … “ part at line 125+) could be expanded and (ii) the size of the embedding should have been ablated in the results section.
* The results section could have been expanded e.g. with (i) the ablation noted above, (ii) extra ablations where, e.g., the feature matching part is turned off.

**Questions:**

* Figure 5 seems to show a case where the transiency is wrong … should the horse really be transient?

**Limitations:**

* To some extent, but the paper could benefit from a more clear failure case section.

---

> ### Author Rebuttal · Authors · 2023-08-07
>
> **Thank you for the detailed review and comments. We appreciate your support for our work. The questions will be addressed below.**
>
> ---
> **Comment 1-1**: The definition of “unconstrained images” should be introduced earlier on. \
> **Answer**: We appreciate for the good point. We will add the definition of “unconstrained images” in line 23 where the concept is firstly introduced. \
>  \
> **Comment 1-2**: Similarly, I do not see much point for Figure 1, as features have been used for this type of matching for years. \
> **Answer**: There have been many works that use deep features to find matching points between images. However, there have been no attempts to use feature fields to optimize NeRF and pose jointly. This approach, optimizing the feature field instead of the radiance field, succeeded to tackle unconstrained images, unlike prior works.
>
> ---
> **Comment 2**: The review section has too much focus only on BARF. \
> **Answer**: In lines 242-246 of the main paper, although NoPe-NeRF is the most relevant baseline which succeeded in training unposed-NeRF in the outdoor scene, it requires training data to be successive frames in a video sequence, but phototourism dataset is not. So, we cannot get a performance of NoPe-NeRF. However, we agree with you that we need more comparisons, so we share the quantitative results of GARF below. It fails to estimate the poses like BARF because it uses photometric loss for unconstrained image optimization.
>
> |  Scenes | ​　　PSNR ↑  | ​　　SSIM ↑ | ​　　LPIPS ↓ | Rotation (°) ↓ | Translation ↓ |
> | --- | --- | --- | --- | --- | --- |
> |   |GARF　　Ours|GARF　　Ours|GARF　　Ours|GARF　　Ours|GARF　　Ours
> | Brandenburg Gate | 07.75　　**25.79** | 0.211　　**0.881** | 1.074　　**0.122** | 133.5　　**0.426** | 4.659　　**0.058** |
> | Trevi Fountain | 10.82　　**22.44** | 0.218　　**0.693** | 1.112　　**0.233** | 154.4　　**1.498** | 7.232　　**0.127** |
> | Taj Mahal | 11.71　　**25.05** | 0.268　　**0.839** | 1.033　　**0.203** | 91.13　　**0.619** | 5.442　　**0.183** |
> | Sacre Coeur | 10.53　　**21.10** | 0.204　　**0.791** | 1.084　　**0.160** | 82.54　　**1.516** | 11.64　　**0.224** |
> | Mean | 10.20　　**23.60** | 0.225　　**0.801** | 1.076　　**0.180** | 115.4　　**0.797** | 7.243　　**0.148** |
> ---
> **Comment 3**: Difficulty of notation in the paper. \
> **Answer**: We will find a way to simplify the notations and reflect them in the final version.
>
> ---
> **Comment 4**: Intuition of candidate embedding and ablation study of its size. \
> **Answer**: We added a more detailed explanation about candidate head to Author Rebuttal. And we share the ablation study on candidate embedding size below. You can see that the performance is not much sensitive to the candidate embedding size, however, there is a big difference when the candidate embedding is not used.
>
> |  Size of $\ell^{(c)}_i$ | ​　0  | ​　16 | ​　32 | ​　64
> | --- | --- | --- | --- | --- |
> | Rotation (°) ↓ | 4.358 | 1.225 | 1.057 | **1.047**
> | Translation ↓ | 0.843 | 0.164 | **0.147** | 0.152
> | PSNR ↑ | 21.45 | 23.26 | **23.47** | 23.40
> | SSIM ↑  | 0.745 | **0.800** | 0.799 | **0.800**
> | LPIPS ↓ | 0.294 | **0.181** | 0.183 | 0.183
>
> ---
> **Comment 5**: More ablation studies \
> **Answer**: We agree with you and we have several additional experiments for ablation including the feature matching part. You can see the results at Brandenburg Gate below. As we mentioned in lines 149-152 of the main paper, it is difficult to estimate pose by using photometric loss because of the diverse appearances of images. As expected, UP-NeRF without feature-surrogate fails to estimate the poses.
>
> |  Method | w/o feature-surrogate  | w/o scheduling | Ours
> | --- | --- | --- | --- |
> | Rotation (°) ↓ | ​　　　　　175.3 | ​　　　172.7 | **0.797**
> | Translation ↓ | ​　　　　　3.846 | ​　　　3.796 | **0.148**
> | PSNR ↑ | ​　　　　　12.29 | ​　　　11.33 | **23.60**
> | SSIM ↑  | ​　　　　　0.645 | ​　　　0.607 | **0.801**
> | LPIPS ↓ | ​　　　　　0.666 | ​　　　0.615 | **0.180**
>
>
> ---
> **Comment 6**: Figure 5 seems to show a case where the transiency is wrong … should the horse really be transient? \
> **Answer**: Static objects (non-transient objects) mean objects that are always there, but “the horse statue” in Figure 5 doesn't appear in every image. It is a temporary installation.

---

### Official Review · Reviewer_6T1t · 2023-07-05

**Soundness:** 4 excellent
**Presentation:** 4 excellent
**Contribution:** 3 good
**Rating:** 7
**Confidence:** 4

**Summary:**

The paper proposes a novel method for optimizing NeRF without a pose-prior and on in-the-wild image collections containing transient occluders and varying lightings. The main contributions are four fold. Firstly, the authors propose a candidate head for NeRFs that uses image-level representations via a learned embedding for compensating inaccurate poses during the early optimization. Secondly, learning a view-independent feature field based on DINO features as intermediate representation increases the robustness of the joint optimization wrt Varying lightings, weather and time. To reduce the impact of transient occluders the authors suggest using a separate network that predicts occluder in 2D image level based on the feature maps. Additionally, to achieve higher accuracy in the geometry the authors apply monocular depth supervision on regions without occluders. Experiments are conducted on four scenes of the Phototourism dataset with initial camera poses set to the identity.

**Strengths:**

1) The paper is very nicely presented and easy to read.
2) The paper tackles the challenging and relevant problem of in-the-wild NeRF reconstruction without given posen. The contribution is clear and solves different subproblems that emerge in this domain, e.g. transient occluders.
3) Most parts are well motivated, the methodology is technical sound and experimentally justified e.g.:
 - Candidate head sounds plausible and improves pose optimization and image quality significantly, see Figure 4 and Table 3.
 - Feature field optimization seems to help for in-the-wild images, see Table 3.
 - Depth supervision improves performance, see Table 3 and Figure 5.
4) The experimental section contains a comparison to BARF, adapted variants with additional supervision cues and numbers for NeRF-in-the-wild. It appears to be a plausible choice and supports the contribution of the paper.
5) The authors provide code in the supplementary materials.


**Weaknesses:**

1) The candidate head is introduced to output color and density, equation 4, however it is later used to predict features, see Figure 2. This appears to be misleading for the readers in the beginning.
2) One of the main limitations of NeRF based methods is the optimization time and an analysis is missing on that. It would be good to discuss the optimization time for the method and the baselines. To get a sense if the time and computational effort is comparable to COLMAP + NeRF-W and others.


**Questions:**

1) The explanation of the loss schedule applied to the candidate head training makes sense however there is no ablation study. What would happen if the weight is not reduced? Why should there be a negative impact if the loss?
2) Can you explain the z(t) in equation 1?
3) Would it make sense to introduce a candidate head with feature output in equation 4?
4) Please provide evidence about the computation time, see Weakness 2.


**Limitations:**

Limitations are discussed in the supplementary.
I'd suggest that the authors discuss the overall optimization time as a general limitation of the method, if applicable. There are state-of-the-art NeRF architecture, such as Instant-NGP that facilitate optimization in a few minutes instead of hour or days, which might be a good follow up.

---

> ### Author Rebuttal · Authors · 2023-08-07
>
> **Comment 1**: Mismatch between Equation 4 and Figure 2. \
> **Answer**:  In lines 158-159, we mentioned the loss $\mathcal{L}_{\text{rgb}}$ is replaced with the loss $\mathcal{L}_{\text{feat}}$, but readers can be confused because feature-surrogate bundle adjustment (Sec 3.2) was described after candidate head (Sec 3.1). We will revise the text (e.g. reorder the sections) so that it can be delivered more clearly.
>
> ---
> **Comment 2-1**: Analysis of computation time between baseline and UP-NeRF.\
> **Answer**: Here, we share a comparison of training time including preprocessing time. Our method requires a relatively shorter training time. Also, NeRF-W needs to obtain camera poses with COLMAP, and it takes over 25 hours as a preprocessing step. However, 10 minutes is enough for UP-NeRF (ours) to prepare DINO features and DPT mono-depth predictions for training.
>
> |  | NeRF-W | | UP-NeRF | |
> | --- | --- | --- | :---: | --- |
> |  | Preprocessing (COLMAP) | Training | Preprocessing (DINO&DPT) | Training |
> | Brandenburg Gate | 29h 29m | 50h 30m | 10m | 46h 30m|
> | Sacre Coeur | 25h 27m | 50h 30m | 10m | 46h 30m|
>
> **Comment 2-2**: Long optimization time as a general limitation of NeRF. Incorporate the proposed method into faster models such as InstantNGP as a future direction.\
> **Answer**: That is a great suggestion. It’ll be good future work. UP-NeRF only requires about 30 minutes of preprocessing time, so using faster models such as Instant-NGP will greatly reduce the training time. According to our quick research, Instant-NGP has not been proven effective on in-the-wild datasets yet. We believe Tackling the challenge and applying the proposed method will lead to interesting technical contributions.
>
> ---
> **Comment 3**: Ablation study of loss scheduling \
> **Answer**: In our final loss Eq. (16), when the schedule is crucial for the initial pose optimization because of color inconsistency. We provide the ablation study of loss scheduling at Brandenburg Gate, and the results table is below. Without loss scheduling, a significant degradation was observed.
>
> |Method |w/o scheduling|Ours|
> | --- | --- | --- |
> |Rotation (°) ↓|172.7|**0.797**|
> |Translation ↓|3.796|**0.148**|
> |PSNR ↑|11.33|**23.60**|
> |SSIM ↑|0.607|**0.801**|
> |LPIPS ↓|0.615|**0.180**|
>
> If you want more explanation of candidate head, please refer to the Author Rebuttal.
>
> ---
> **Comment 4**: Explanation of $\mathbf{z}(t)$ in equation 1. \
> **Answer**: We apologize we didn’t explain $\mathbf{z}(t)$ notation properly. $\mathbf{z}(t)$ is the intermediate feature of MLP, which is just an output of MLP $\theta_1$. It will pass through both MLPs $\theta_2$, $\theta_1$.
>
> ---
> **Comment 5**: Would it make sense to introduce a candidate head with feature output in equation 4?\
> **Answer**: The MLP parameterized by $\theta_1$ a base network shared by two heads: shared head and candidate head. The feature $\mathbf{z}(t)$ is output of MLP $\theta_1$ and it is transformed the positional information. The feature (or transformed position embedding) was fed into MLP theta 3 as input.

---

> > ### Comment · Reviewer_6T1t · 2023-08-20
> >
> > Thanks for providing detailed explanation. After reading all reviews and the rebuttal, I keep my score and recommend to accept the paper.

---

### Official Review · Reviewer_7yJa · 2023-07-06

**Soundness:** 3 good
**Presentation:** 3 good
**Contribution:** 3 good
**Rating:** 7
**Confidence:** 4

**Summary:**

The paper proposes a joint camera pose and NeRF optimisation method that can handle transient scenes, including moving objects and various light conditions, by integrating NeRF-W, BARF, NoPe-NeRF, and DINO-based feature-metric loss in a sophisticated way.

The method is evaluated on four scenes in the Phototourism dataset, showing good performance compared with (modified) BARF baselines.

---
**After rebuttal**: I have read authors' rebuttal and it addresses my concerns.

**Strengths:**

It’s a challenging task to estimate camera poses in scenes with moving objects and different light conditions, especially in a NeRF setup. This method proposes to make the pipeline more robust by considering
* Moving objects and lighting conditions similar to NeRF-W;
* Pose optimisation similar to BARF;
* Monocular depth un-distortion similar to NoPe-NeRF; and
* DINO-based Feature-metric.

In short, the proposed method is novel and leads to promising results. It’s also a plus that code is also provided in supplementary.

**Weaknesses:**

I think a primary concern is how robust the pose estimation is in more scenes. The method is only evaluated in 4 scenes. Is it possible to have an experiment, especially for pose estimation successful rate on more scenes?

I understand that it takes a long time for NeRF to reach the best rendering quality, but we should be able to tell if the pose optimisation is successful far before NeRF converges. For example, we can consider pose estimation successful if rotation error is lower than $\theta$ degrees in $x$ epochs. In this way, we can see the pose estimation success rate in many more scenes.

**Questions:**

See the weakness section.

**Limitations:**

Yes

---

> ### Author Rebuttal · Authors · 2023-08-07
>
> **Thank you for the constructive comments, including new experimental settings. We address the reviewer's questions below.**
>
> ---
> **Comment 1**: Experiments in more scenes.
>
> **Answer**:
> We provide more experimental results. As requested, we set an error criterion of 20 degrees, which was used by NeRF-- [3] to determine whether the pose estimation was successful or not . And at 10% of the total iterations, it was measured whether it was lower than the error criterion. The result is provided in Fig. 2 of the PDF. The experiments demonstrate that our method successfully learns pose estimation (< 20 degree) in all five scenes below before 10% of the total iterations.
>
> - **3 more scenes in phototourism dataset.** Since NeRF-W did not release pre-processed images of all scenes, we evaluated the model only in four scenes (Brandenburg Gate, Taj Mahal, Trevi Fountain, and Sacre Coeur) on the phototourism dataset in the main paper. We implemented the preprocessing step as similarly as possible and evaluated three more scenes (Buckingham Palace, Pantheon Exterior, Nara Temple).
> - **2 more scenes in outdoor image dataset.** We additionally presented a nice outdoor image dataset with color variation and transient occluders as in NeRF-OSR [2].
>
> ### References
> [1] NIMA: Neural image assessment. IEEE TIP, 2018.
>
> [2] NeRF-OSR: Neural Radiance Fields for Outdoor Scene Relighting. ECCV 2022.
>
> [3] NeRF−−: Neural radiance fields without known camera parameters. arXiv preprint arXiv:2102.07064,338 2021.

---

> > ### Comment · Reviewer_7yJa · 2023-08-16
> > **thanks for additional results**
> >
> > Thanks for providing more results and addressing my concerns. I'll raise my final rating.

---

### Author Rebuttal · Authors · 2023-08-07

Multiple reviewers asked about our Candidate Head. We address the common comment below.

Q. **Additional explanation of Candidate Head**

During the initial phases of joint training for NeRF and camera pose estimation, NeRF struggles to accurately capture intricate scene details. This limitation is particularly pronounced in the case where images are hard to infer their pose based solely on coarse information. We refer to these images as "hard-pose images." These inaccurately aligned images subsequently hinder the NeRF training, introducing erroneous supervision.
To mitigate this problem, we proposed a novel architecture with two distinct NeRF heads: the shared head and the candidate head. The intuition is to consider easy/distinct pieces first to complete a large jigsaw puzzle and handle hard pieces later. The shared head is responsible for generating essential parameters, namely density $\sigma$ and color $\hat{\mathbf{c}}$, which are shared across all images. On the other hand, the candidate head generates unique parameters, including density $\sigma^{(c)}_i$ and color $\hat{\mathbf{c}}^{(c)}_i$, tailored to each individual image. In the initial stages of training, the shared head is predominantly learned by easy-pose images. Meanwhile, the candidate head is primarily trained by hard-pose images. This remedy ensures that the candidate head effectively prevents the shared head from being influenced by incorrect guidance. As the shared head becomes accurate enough to facilitate the pose estimation of hard-pose images, these challenging instances are gradually incorporated into the shared head by our loss scheduling that adjusts the weights between two heads, i.e. $w_s$$_e$ in Equation 7 of the main paper.

---

### Decision · Program_Chairs · 2023-09-21

**Decision:**

Accept (poster)

**Comment:**

All the reviewers recommend the acceptance of the work. Reviewers appreciated the well-written paper with good technical contributions. Reviewers raised several clarification questions and also raised concerns related to results on only one dataset. Authors addressed several of these questions in their responses. The reviewers did raise some valuable concerns and suggestions that should be addressed in the final camera-ready version of the paper, which include adding the relevant rebuttal discussions and revisions in the main paper. The authors are encouraged to make the necessary changes to the best of their ability.